# One-Year Post-Vaccination Longitudinal Follow-Up of Quantitative SARS-CoV-2 Anti-Spike Total Antibodies in Health Care Professionals and Evaluation of Correlation with Surrogate Neutralization Test

**DOI:** 10.3390/vaccines11020355

**Published:** 2023-02-03

**Authors:** Yesim Tuyji Tok, Rabia Can Sarinoglu, Seyhan Ordekci, Serife Yilmaz, Gunes Ozcolpan, Aysen Bayram, Okan Kadir Nohut, Ipek Kocer, Ufuk Hasdemir, Mert Ahmet Kuskucu, Dildar Konukoglu, Aysegul Gozalan, Kenan Midilli, Gulden Celik

**Affiliations:** 1Department of Medical Microbiology, Istanbul University-Cerrahpasa, Cerrahpasa Medical School, 34098 Istanbul, Turkey; 2Department of Medical Microbiology, Bahcesehir University Medical School, 34353 Istanbul, Turkey; 3Department of Medical Microbiology, Ministry of Health University Mehmet Akif Ersoy Thoracic and Cardiovascular Surgery Training and Research Hospital, 34303 Istanbul, Turkey; 4Department of Medical Microbiology, Karabuk University Medical School, 78100 Karabük, Turkey; 5Department of Medical Microbiology, Aydın Adnan Menderes Medical School, 09100 Aydın, Turkey; 6Department of Medical Microbiology, SANKO University Medical School, 27090 Gaziantep, Turkey; 7Fikret Biyal Central Research Laboratory, Istanbul University-Cerrahpasa, Cerrahpasa Medical School, 34098 Istanbul, Turkey; 8Department of Medical Microbiology, Marmara University Pendik Training and Research Hospital, 34854 Istanbul, Turkey; 9Department of Medical Biochemistry, Istanbul University-Cerrahpasa, Cerrahpasa Medical School, 34098 Istanbul, Turkey; 10Department of Medical Microbiology, Alanya Alaaddin Keykubat Medical School, 07400 Alanya, Turkey

**Keywords:** COVID-19, heterologous, inactivated, mRNA, vaccine

## Abstract

Numerous vaccines have been generated to decrease the morbidity and mortality of COVID-19. This study aims to evaluate the immunogenicity of the heterologous boosts by BioNTech against homologous boosts by CoronaVac at three-month intervals in two health care worker (HCW) cohorts, with or without prior COVID-19, for one year post-vaccination. This is a prospective cohort study in which the humoral responses of 386 HCWs were followed-up longitudinally in six main groups according to their previous COVID-19 exposure and vaccination status. Anti-SARS-CoV-2 spike-RBD total antibody levels were measured and SARS-CoV-2 neutralization antibody (NAbs) responses against the ancestral Wuhan and the Omicron variant were evaluated comparatively using international standard serum for Wuhan and Omicron, as well as with the aid of a conversion tool. The anti-SARS-CoV-2 spike-RBD total Ab and Nab difference between with and without prior COVID-19, three months after two-dose primary vaccination with CoronaVac, was statistically significant (*p* = 0.001). In the subsequent follow-ups, this difference was not observed between the groups. Those previously infected (PI) and non-previously infected (NPI) groups receiving BioNTech as the third dose had higher anti-SARS-CoV-2 spike total Ab levels (14.2-fold and 17.4-fold, respectively, *p* = 0.001) and Nab responses (against Wuhan and Omicron) than those receiving CoronaVac. Ab responses after booster vaccination decreased significantly in all groups at the ninth-month follow-up (*p* < 0.05); however, Abs were still higher in all booster received groups than that in the primary vaccination. Abs were above the protective level at the twelfth-month measurement in the entire of the second BioNTech received group as the fourth dose of vaccination. In the one-year follow-up period, the increased incidence of COVID-19 in the groups vaccinated with two or three doses of CoronaVac compared with the groups vaccinated with BioNTech as a booster suggested that continuing the heterologous CoronaVac/BioNTech vaccination, revised according to current SARS-CoV-2 variants and with at least a six-month interval booster would be an effective and safe strategy for protection against COVID-19, particularly in health care workers.

## 1. Introduction

The ongoing pandemic caused by severe acute respiratory syndrome coronavirus-2 (SARS-CoV-2) has promoted the most rapid vaccine technology development in the history of science, worldwide. New anti-SARS-CoV-2 vaccine development platforms using the inactivated virus, nonreplicating viral vector, subunit, viral-like particle, DNA, or mRNA, were implemented [1,2]. Fourteen vaccines were accepted as reliable, immunogenic, and effective against SARS-CoV-2 to be used by the general population for reducing the spread of the disease, critical cases, and deaths in different countries, thus far. However, with the worldwide spread of Omicron and its later subvariants, the issue of which type of COVID-19 vaccine protects for how long has come to the fore. The vaccination program against COVID-19 in Turkey started in February 2021 with the two-dose CoronaVac vaccine. CoronaVac^®^ is an inactivated SARS-CoV-2 vaccine developed by Sinovac Life Sciences Co., Ltd. (Beijing, China) and is among the current vaccines approved by the WHO to combat COVID-19 [3,4]. In Turkey, about fifty-eight million people have received the first vaccine dose, and 64% of the target population was fully vaccinated until December 2022. In July 2021, Turkey also imported BioNTech mRNA COVID-19 vaccine (Pfizer-BioNTech, Chesterfield, MO, USA) and health care workers were offered CoronaVac or BioNTech vaccine option as a booster dose. The use of booster doses was approved in Turkey for high-risk populations and subjects with more than five months after the second dose applied in a 0–28-day vaccination schedule [5]. However, the interval between the doses may vary depending on whether individuals have COVID-19 or general indications for vaccine postponement during that interval. It is also possible to mix and match vaccines in specific situations such as the availability of a vaccine or adverse reactions following vaccine administration. As a result, there have been variations in terms of vaccination schedules and types, and therefore in terms of immune status against COVID-19 in Turkey.

Serological assays are essential in the detection of anti-SARS-CoV-2 antibodies (Abs) and monitoring convalescent and vaccine-induced immunity [6]. Mainly, they play a supplementary but indispensable role in diagnosing suspected cases with a negative swab, in asymptomatic or past SARS-CoV-2 infection patients, for epidemiological assessment, assuring the contact-tracing of positive cases and identifying new foci, and in evaluating vaccine efficacy and therapeutic antibody development [6]. On the other hand, serum neutralizing antibody (NAb) titers are thought to be highly predictive of immune protection [7]. Antibody titers decrease in time after immunization; however, the length of this protective immunity is still unclear [8,9]. Besides this, antibodies have lower levels of neutralization against highly transmissible variants of the virus as compared with the original vaccine strain. This is potentially decreasing the effectiveness of these vaccines as new variants emerge [10].

Nowadays, the gold standard to evaluate humoral immunity for SARS-CoV-2 is the plaque reduction neutralization test (PRNT). However, these assays use SARS-CoV-2 replicating virus. PRNTs are unfavorable in that they require BioSafety Level (BSL)-3 laboratories and often require intense and time consuming (about a week) work by skilled operators, as well as not being readily amenable to automatization [11]. Due to its complex process, more practical and standardized tests have been validated, such as the cPass™SARS-CoV-2 Neutralization Ab detection kit (GenScript, Piscataway, NJ, USA), which demonstrated high sensibility and specificity compared with PRNT. Likewise, the Elecsys^®^ anti-SARS-CoV-2 S test (Roche Diagnostics GmbH, Mannheim, Germany) has shown a high correlation with Ab neutralization cPass™ and a moderate correlation compared with PRNT. The Food and Drug Administration (FDA) has approved both tests [12].

While neutralizing antibody level assessment is known to be a good biomarker for the correlation of protection against SARS-CoV-2 infection, many results from these studies are presented using assays that have not been calibrated using an international standard unit (IU), making it difficult to define the exact level of neutralizing antibodies required for protection and to compare with current and future studies.

This study aims to assess and provide large cohort data on the immunogenicity of heterologous or homologous boosters after a two-dose inactivated vaccine followed by BioNTech mRNA vaccine or CoronaVac vaccine among Turkish health care workers (HCWs) at four different sampling time points during one year post-vaccination.

## 2. Materials and Methods

### 2.1. Participants and Study Design

HCWs who presented evidence of the primary vaccination were invited to participate in this prospective cohort study, with a first sampling 3 months after the second dose of CoronaVac, in June 2021 (Figure 1). The primary vaccination regimen adopted was two doses of CoronaVac with a time interval of 28 days. Firstly, two main groups were formed from HCWs who were documented to have had a previous infection (PI) by COVID-19 RT-PCR which is considered the gold standard in diagnosis (PI group n = 201), and who did not have COVID-19 (NPI group n = 185). Then, by the 6th month sampling in September 2021, the participants were analyzed in six subgroups according to whether they were vaccinated with BioNTech/CoronaVac as a booster dose, after primary vaccination with two-dose of CoronaVac, or not having any booster (PIBB = previously infected BioNTech booster receiver, PICB = previously infected CoronaVac booster receiver, PINB = previously infected non booster receiver, NPIBB = non-previously infected BioNTech booster receiver, NPICB = non-previously infected CoronaVac booster receiver, and NPINB = non-previously infected non booster receiver groups) (Figure 1). Whole venous blood samples (5 mL) were collected in vacuum tubes without anticoagulant. With the first serum sampling beginning 3 months after two doses of CoronaVac, serum anti-SARS-CoV-2 spike total antibodies were longitudinally followed-up every three months at the six hospitals providing tertiary health care in different regions of Turkey between March 2021 and February 2022. Only participants who agreed to give serum samples at four time points, every 3 months during the one-year study, were included in the study (Figure 1). Informed consent was obtained from all volunteers upon enrollment. The exclusion criteria at enrollment were ongoing immunosuppressive medication, any vaccination within one-month, and having received any blood components or intravenous immunoglobulin within three months. The termination criteria were either SARS-CoV-2 infection or having received unscheduled vaccination against COVID-19 which could generate bias when comparing titers of antibodies.

### 2.2. Demographics and Clinical Data

Baseline demographics and clinical data, such as medical history, current medications used, history of exposure to COVID-19 patients, and history of SARS-CoV-2 infection and vaccination were collected retrospectively from electronic medical records and based on participants’ statements (Table 1). During the study period, participants who reported symptoms or contact with infected individuals were also applied molecular tests (qRT-PCR) and a clinical evaluation.

### 2.3. Assessment of Anti-SARS-CoV-2 Spike Total Antibodies

The serum was separated by centrifugation. Investigation of anti-SARS-CoV-2 spike-RBD total antibodies was performed using a quantitative electrochemiluminescence immunoassay (ECLIA) by Elecsys^®^ SARS-CoV-2 S (Roche Diagnostics, Basel, Switzerland), following the manufacturer’s recommendations, using Cobas e602 immunoassay analyzer (Roche Diagnostics, Rotkreuz, Switzerland). Elecsys^®^ Anti-SARS-CoV-2 S assay uses a recombinant protein representing the RBD of the spike antigen in a double-antigen sandwich assay format. The analytical measuring interval was 0.40–250 U/mL. A volume of 200 mL of 100-fold diluted sample was used following the manufacturer’s protocol. For the participants who had anti-SARS-CoV-2 spike total antibodies over the maximum measuring range (>25,000 U/mL), the samples were further diluted to 1000-fold and re-evaluated using an Elecsys^®^ Diluent Universal kit. The assigned U/mL is equivalent to binding antibody units (BAU/mL), as defined by the WHO International Standard for anti-SARS-CoV-2 immunoglobulin (NIBSC code 20/136) (Results reported in U/mL do not need to be converted to another unit and can be directly compared to the results of other studies using BAU/mL). Numeric values were interpreted as “negative” (<0.80 U/mL) and as “positive” (≥ 0.80 U/mL).

### 2.4. Assessment of SARS-CoV-2 Neutralizing Antibodies to the Ancestral Wuhan and Omicron Variant

We evaluated the circulating NAbs in 60 of the volunteers at four different time points using surrogate virus neutralization tests (sVNT). The virus neutralization was tested against the ancestral Wuhan strain and the currently dominant VOC (variant of concern), Omicron, with a competitive ELISA-based sVNT: the cPass™SARS-CoV-2 Neutralization Antibody Detection Kit (GenScript Biotech, Piscataway, NJ, USA). Serum samples were diluted 1:10 with buffer and incubated with horse-radish peroxidase-conjugated RBD for 30 min at 37 °C. Next, 100 µL of the sample mixture was added to a capture plate pre-coated with human angiotensin-converting enzyme 2 (ACE2) and incubated for 15 min at 37 °C. After washing, 100 µL of 3,3′,5,5′-tetramethylbenzidine (TMB) chromogen solution was added, and the plate was incubated in the dark for 15 min at room temperature. After the addition of a 50 µL stop solution, the samples were read at 450 nm. If the patients developed NAbs, the interaction of RBD-ACE2 was interrupted, provoking a signal lost from the HRP. At the same time, if there was an absence of NAbs in the sample, the HRP-RBD would bind with the ACE2 and generate a colorimetric signal. The percentage of signal inhibition (PSI) was determined by subtracting one minus the division of the sample’s optical densities (OD) with the OD of the negative control, multiplied by 100. A PSI bigger or equal to 30% was considered positive. The seroconversion rate was defined as sVNT ≥68%, which was adopted from the US FDA’s guidelines for a high titer of COVID-19 convalescent plasma [13]. A standard curve was used to plot the neutralization response in the samples as international units (IU) by using serial ten-fold dilutions of the WHO International Standard for SARS-CoV-2 Wuhan and Omicron variant antibody, which was prepared according to the manufacturer’s instructions [14]. PSI values were also converted to IU/mL using a conversion tool [15], and the agreement between the PSI values of all samples and IU/mL equivalents obtained using the international standard serum and the conversion tool was evaluated.

### 2.5. Statistical Analysis

Statistical analysis was performed in SPSS V21.0 software and Prism V 9.2.0 (Graphpad Software, LLC, San Diego, CA, USA). The categorical variables were presented as absolute and relative frequency, and the symmetrically distributed variables were reported as the mean and standard deviation (±Std). As suggested by the WHO, guidelines on clinical evaluation of antibody titers were reported as geometric means (GMT) [16]. We used the Mann–Whitney U statistical test to compare the quantitative variables since its distribution was not symmetrical. Statistical differences for the anti-SARS-CoV-2 spike total Abs results of the six groups were considered by one-way ANOVAs analysis on the logarithms of the data. Symmetrically distributed serum NAb values were compared with the Student’s *t*-test among the PIBB, PICB, NPIBB, and NPICB groups for four sampling periods. Friedman test was used to compare the change of Ab values over the sample periods of each study group. The correlation between anti-SARS-CoV-2 spike total Abs and NAbs was evaluated with linear regression analysis and the results were presented with the 95% confidence interval (CI). The values of *p* < 0.05 and R^2^ = 0.5–1 were considered statistically significant.

## 3. Results

Of the HCWs vaccinated with two doses of CoronaVac, 386 consented to participate and were selected for the study. One hundred seventeen HCWs were excluded from the study during the follow-up period because they were either diagnosed with COVID-19, had received unscheduled vaccination, or could not give periodic serum samples for any reason. Finally, in the twelfth-month period, the Ab results of 269 health care workers who were able to give serum samples at all four time points and were accepted as appropriate for the study procedure were analyzed (Figure 1).

The median age of participating HCWs was 37.04 years (Std = ±8.72), and 227 (58.8%) of them were females. Statistically, no significant gender and age differences were observed between the PI and NPI groups. While 28.1% of the PI group had risky medical conditions for severe COVID-19, this rate was found to be statistically significantly lower in the NPI (19.7%) (*p* = 0.001) (Table 1).

Anti-SARS-CoV-2 spike total Ab titer difference between the PI and NPI groups, three months after primary vaccination with two-dose CoronaVac, was statistically significant (PI = 275.6 U/mL, NPI = 60.7 U/mL, respectively, *p* = 0.001). Additionally, the Nab levels were fivefold higher in the PI group compared with the NPI group (for Wuhan; PI = 148.5 IU/mL, NPI = 30.54 IU/mL, lower-upper 95% CI of geo. Mean = 58.6–135.4, 18.6–30.45 and for Omicron PI = 98.32 IU/mL, NPI = 18.8 IU/mL, lower-upper 95% CI of geo. Mean = 96.22–101.34, 14.62–24.24, respectively, *p* = 0.001) (Table 2). This difference was not observed in the subsequent follow-ups and there was no statistical association between the history of previous SARS-CoV-2 infection and the titer of anti-SARS-CoV-2 spike total Ab or NAbs after the booster.

The changes in anti-SARS-CoV-2 spike total Ab titers in four sampling periods for 12 months are summarized in Figure 2 as log_10_ values. Additionally, neutralizing Ab levels measured by sVNT separately against Wuhan and Omicron variant for the PIBB, PICB, NPIBB, and NPICB groups in the same periods are summarized in Figure 3a,b.

Receiving BioNTech as a booster dose provided significantly higher anti-SARS-CoV-2 spike total Ab levels and NAb responses (against the ancestral Wuhan strain and the Omicron variant) than those receiving a CoronaVac booster at the six-month follow-up. Compared with the 3rd-month measurement, in the non-previously infected BioNTech booster receiver group—NPIBB the increase in anti-SARS-CoV-2 spike total Ab was 17.4-fold higher than that in the NPICB group; in the previously infected BioNTech booster receiver group—PIBB, the increase in anti-SARS-CoV-2 spike total Ab was 14.2-fold higher than that in the PICB group (Figure 2) (NPIBB group = 13,266 U/mL, 422 IU/mL, and 54.47 IU/mL; PIBB group = 11,237 U/mL, 436.8 IU/mL, and 145.4 IU/mL; NPICB group = 759 U/mL, 104.5 IU/mL, and 11.32 IU/mL; PICB group = 787 U/mL, 85.66 IU/mL, and 12.87 IU/mL, anti S Ab, NAb_Wuhan_, and Nab_Omicron_ GMTs, respectively, *p* = 0.001) (Table 2). In total, 96.66% of the booster receiver groups (58/60) displayed a positive presence of neutralizing antibodies; however, seroconversion (defined as sVNT ≥ 68%, according to the manufacturer’s instructions) occurred in 100% (30/30) of the group that received BioNTech booster, while this rate remained at only 26.6% (8/30) in the CoronaVac booster group. In the 6-month measurement, anti-SARS-CoV-2 spike total Abs were below the protective level in the groups not having any booster vaccine after primary vaccination with two-dose CoronaVac (NPINB group = 275.6 U/mL and PINB group = 234.0 U/mL) (Table 2) [16].

Anti-SARS-CoV-2 spike total Abs decreased significantly in all groups at the 9-month follow-up (to about the half titers; NPIBB = 6261 U/mL, PIBB = 5281 U/mL, NPICB = 356 U/mL, and PICB = 475 U/mL, *p* < 0.005) (Table 2). Anti-SARS-CoV-2 spike total Ab GMTs in NPIBB were 17.5-fold higher than that of the NPICB group and in PIBB were 11.1-fold higher than that of the PICB group (Figure 2); NAb GMTs were 5.1-fold higher in the NPIBB than the NPICB group and were 6.4-fold higher in the PIBB than the PICB group (Figure 3a,b). Nevertheless, it was still higher in the all booster received groups than the post-primary vaccination levels. On the other hand, the anti-SARS-CoV-2 spike total Ab GMTs were 64.1 U/mL in the NPINB group and 85.8 U/mL in the PINB group.

In the 12-month follow-up measurement, anti-SARS-CoV-2 spike total Abs were above the protective level in the both NPIBB and PIBB groups and were increased more than twofold by the second BioNTech booster compared with the previous sampling period (anti-SARS-CoV-2 spike total Abs were increased from 6261 U/mL to 13,105 U/mL in the NPIBB group and from 5281 U/mL to 12,705 U/mL in the PIBB group; NAbs for Wuhan were increased from 363 IU/mL to 412 IU/mL in the NPIBB group and from 322 IU/mL to 429 IU/mL in the PIBB group; NAbs for Omicron were increased from 95.1 IU/mL to 206 IU/mL in the NPIBB group and from 118 IU/mL to 210 IU/mL in the PIBB group) (Table 2).

We further observed that, when comparing the NPI groups (in NPIBB—42/94, 44.6%; in NPICB—10/14, 71.4%; and in NPINB—5/8, 62.5.8%) with the PI groups (in PIBB—21/101, 20.8%; in PICB—19/26, 73.0%; and in PINB—16/22, 73.0%), fewer patients acquired COVID-19 in the BioNTech booster receiver groups (NPIBB and PIBB) in the follow-up of four sampling periods after vaccination *(p* < 0.05) (Figure 1).

In the groups that received the second BioNTech booster (the fourth dose of vaccination) six months after the first BioNTech, the maximum Ab level detected was 64.101 U/mL. Additionally, the maximum titers detected in the HCWs who received the fifth dose (3rd BioNTech) were still not extremely high (max = 62,102 U/mL) (Figure 1).

In the 9th month analysis, those who had COVID-19 during this period, those who received a second booster, or those who did not receive a booster before and received their first booster during this period were not included. In the 12th month analysis, the arm that received the fourth dose of BioNTech booster and the main arms remaining on second and third doses of CoronaVac vaccine during this period were included. Participants who had COVID-19 during this period or those who did not receive any booster before and received their first booster during this period were excluded from the 12th month analysis.

The data from 60 selected participants’ NAbs were compared in four groups who received first and second BioNTech or CoronaVac booster, with or without previous COVID-19 (PIBB, PICB, NPIBB, and NPICB). Those who did not receive any boosters after the two-dose CoronaVac, those who did not receive a second booster yet in the 12th month sampling, those who received a third booster, or those who had COVID-19 between the sampling periods were not included in the sVNT.

Neutralizing Ab levels from sVNT for both Wuhan and Omicron showed a strong correlation with anti-SARS-CoV-2 spike total Ab at all four follow-up periods (*p* = 0.001; r^2^ > 0.89). When we performed linear regression analysis between Roche ECLIA and sVNT results, among the percent inhibition values, conversion tool, and using an international standard, the best correlation with ECLIA was obtained by sVNT when international standards for Wuhan and Omicron were used (Figure 4).

## 4. Discussion

In this prospective cohort study, we periodically evaluated the humoral responses of 386 health care workers to SARS-CoV-2, starting with the serum anti-SARS-CoV-2 spike total and NAb measurement 3 months after two doses of CoronaVac administration as the primary vaccination. We found a strong humoral response at 60–90 days in the BioNTech booster receiver group. A seroconversion (PSI ≥ 68%) was achieved in all PIBB and NPIBB group participants. However, this humoral response gradually decreased until the ninth month sampling period, especially in the CoronaVac booster group, where anti-SARS-CoV-2 spike total and NAbs were lower than those of the BioNTech booster group. Considering anti-SARS-CoV-2 spike total Ab levels, seroconversion percentage and reinfection rates, the third dose of CoronaVac as a booster did not provide an adequately inclusive humoral response. These findings were lower than those reported in phase studies of CoronaVac clinical trials [17], although were consistent with another study [18].

No statistical relationship was found between gender and pre- or post-vaccination COVID-19 occurrence. However, we did not find a difference in the Ab levels according to age group, probably because most of our participants were adults aged 30–50 years, and these differences in the humoral response are primarily observed in population-based studies involving older decades. As expected, the incidence of contracting the disease was increased in people with risky medical conditions such as diabetes, hypertension, pulmonary disease, immune deficiencies, obesity, or smoking history.

Comparing the PI and NPI groups after primary vaccination with two doses CoronaVac, the fivefold difference in the levels of anti-SARS-CoV-2 spike total and NAb showed that the humoral response obtained with two doses of CoronaVac was much less than the immune response obtained with a natural infection process or with a single dose of the BioNTech vaccine [18,19].

The findings of our study showed that Anti-SARS-CoV-2 spike total Ab levels reached high protective levels after the first BioNTech booster, regardless of previous COVID-19 exposure, in both PI and NPI groups. This result was also seen as values close to 100% neutralization in NAb levels. Consistent with previous studies, the BioNTech booster vaccine after primary vaccination with two doses of CoronaVac produces a stronger humoral immune response than that of the CoronaVac booster (third CoronaVac). However, whether the booster is administered with BioNTech or CoronaVac, the response in both is determined above the protective level and the primary vaccination response. In a recent study, it was observed that neutralizing antibodies increased ten-fold compared with their previous results after CoronaVac booster administration, and in another study, this rate was found to be significantly higher, such as eight hundred-fold, with BioNTech booster administration [20,21]. Similar findings have been reported in patients who received a homologous or heterologous booster after two doses of CoronaVac. Additionally, Costa C. et al. studied comparing vaccines of different designs and booster applications after two doses of CoronaVac, and reported that the immune response provided by all vaccines was higher than that of the primary vaccination, but the most effective response was obtained with the BioNTech booster [22].

Although Ab levels tended to decrease gradually 6 months after the BioNTech or CoronaVac booster, and were still higher than the primary vaccination Ab levels in both groups, there was a substantial difference of more than 10-fold difference in favor of BioNTech, consistent with other studies [20,21,23]. Additionally, there was a significant difference between those who received the BioNTech booster and those who received the CoronaVac booster in terms of re-infection or breakthrough infection. Despite previous studies in which the BioNTech booster was found to be very effective in terms of reducing COVID-19 reinfection and mortality rates, these immune responses have decreased against emerging variants/subvariants that are now common due to the rapid evolution of the virus [24,25,26]. However, it has been shown that equivalent neutralizing immune responses of more than 70% can be obtained with a second BioNTech booster administered, intervally [27]. Our study ascertained the presence of circulating neutralizing antibodies, employing the cPass™ SARS-CoV-2 Neutralization Antibody Detection Kit, which had demonstrated a high sensibility, specificity, and strong correlation with the gold standard, PRNT [16]. The first peak of neutralizing antibodies was observed during the booster vaccination, in which 96.66% of the booster receiver groups displayed a positive presence of neutralizing antibodies, however, seroconversion occurred in 100% of the group that received the BioNTech booster, while this rate remained at only 26.6% in the CoronaVac booster. This proportion was inferior to the data reported in phases 1 and 2 of the CoronaVac clinical trials [17]. The second peak of neutralizing antibodies was observed after the second BioNTech booster in the twelfth month control, and seroconversion reached 100%, all over. Nevertheless, NAb responses to Omicron in all four sample periods were parallel to NAb against ancestral Wuhan, but mean titers were found approximately 50% lower. This indicates that people become susceptible to Omicron and its subvariants, despite the booster doses, unless vaccines are revised for emerging variants.

Previously, clinical trials were conducted in which various vaccines were mix–matched and the immune responses obtained by booster vaccine applications and vaccine safety were investigated [28]. In our study, after BioNTech administration as the fourth dose of vaccination (second booster dose), no extremely high Ab level was detected, which would cause immune response irregularity. No higher result was found even in those who received BioNTech as the fifth dose vaccine. This suggested that BioNTech booster administration is safe, but boosters more frequently than at 6-month intervals do not bring any additional benefit.

A limitation of our study is that our results cannot be attributed to the entire the population, as the study was designed to include HCWs, and our cohort consisted of middle-aged individuals without serious health conditions. Additionally, the pre-vaccination baseline humoral immune response status of the participants could not be evaluated. It is relevant to mention that although the surrogate virus neutralization test target receptor-binding domain (RBD) is the major binding region for neutralizing antibodies, other regions, such as the N-terminal domain (NTD) of viral spike protein 1 (S1), also bind to neutralizing antibodies [29]. On the other hand, humoral immunity is not the only protective response against SARS-CoV-2. It has been observed that patients vaccinated with CoronaVac who responded with NAbs had a significantly higher level of anti-SARS-CoV-2 spike IgG and a tendency to generate more spike-specific memory B cells than non-responders. They also showed similar titers of spike-specific memory CD8 T cells and CD4 T cells compared with convalescent patients. Furthermore, this adaptive immunity has been associated with a reduction in disease severity. In our study, only humoral immunity could be evaluated, not cellular immunity, which is well known to have an important role in the immune response to viral infections.

## 5. Conclusions

Currently, few studies have evaluated the immune response of homologous or heterologous-BioNTech boosting after two doses of CoronaVac. In this study, the humoral response with heterologous BioNTech or homolog CoronaVac boosting after two doses of CoronaVac in detail up to 12 months has been followed up. A heterologous booster of the BioNTech mRNA vaccine, after three months of two doses of the inactivated vaccine against SARS-CoV-2, CoronaVac, induced a robust immune response, regardless of the history of previous infection or not.

Surrogate neutralization assays calibrated to the WHO international standard, such as cPass, represent a good first step towards such an international harmonization goal. More so than ever, with the Omicron variant spreading rapidly across the globe, a harmonized approach for the assessment of risk and correlation of protection is highly desirable. A fourth dose (second booster) should be recommended, particularly in health care workers, at least six months later than the first booster dose. Further studies of the second booster-immunogenicity with a larger group of participants including T-cell responses are needed.

## Figures and Tables

**Figure 1 vaccines-11-00355-f001:**
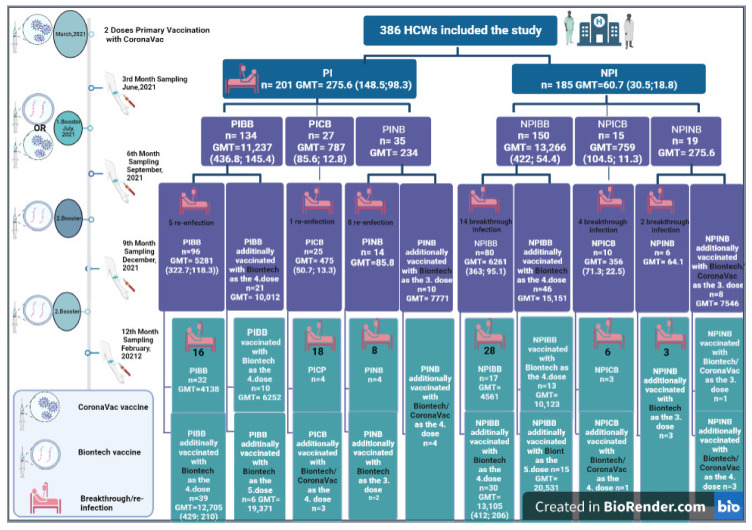
Graphical abstract of the study and the subgroups included (PI = previously infected, NPI = non-previously infected, PIBB = previously infected BioNTech booster receiver, PICB = previously infected CoronaVac booster receiver, PINB = previously infected non booster receiver, NPIBB = non-previously infected BioNTech booster receiver, NPICB = non-previously infected CoronaVac booster receiver, NPINB = non-previously infected non booster receiver groups, n = number of the participants, GMT = geometric mean of anti-SARS-CoV-2 spike total Ab (GMT of NAb_Wuhan_; GMT of NAb_Omicron_).

**Figure 2 vaccines-11-00355-f002:**
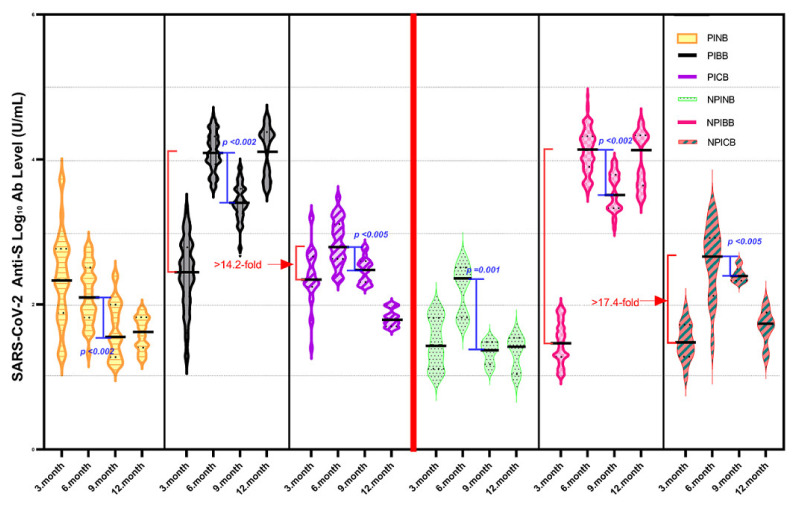
One year comparative evaluation of anti–SARS–CoV–2 spike total Log_10_ Ab levels (U/mL) of PINB, PIBB, PICB, NPINB, NPIBB, and NPICB groups in four sampling periods with 3–month intervals. (PINB = previously infected non booster receiver, PIBB = previously infected BioNTech booster receiver, PICB = previously infected CoronaVac booster receiver, NPINB = non–previously infected non booster receiver, NPIBB = non-previously infected BioNTech booster receiver, and NPICB = non-previously infected CoronaVac booster receiver group).

**Figure 3 vaccines-11-00355-f003:**
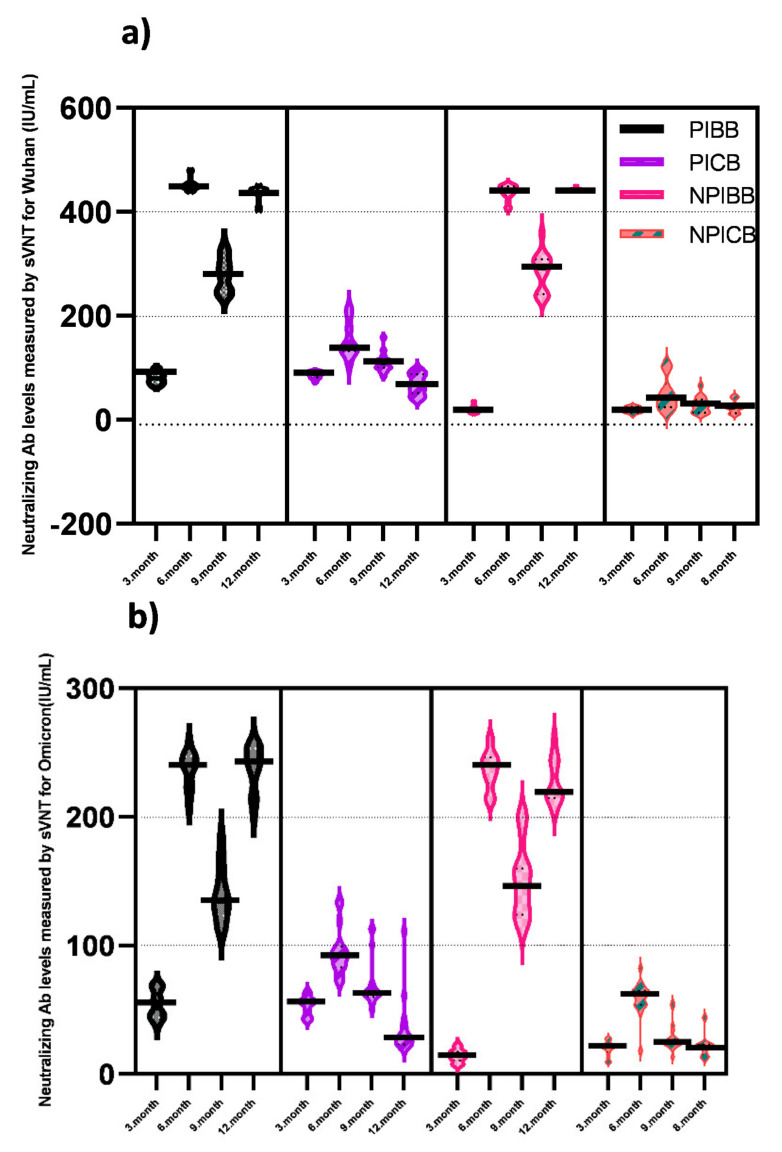
One year comparative evaluation of SARS-CoV-2 Neutralizing Ab levels (IU/mL) measured by sVNT against (**a**) Wuhan and (**b**) Omicron of PIBB, PICB, NPIBB, and NPICB groups in four sampling periods with 3-month intervals.

**Figure 4 vaccines-11-00355-f004:**
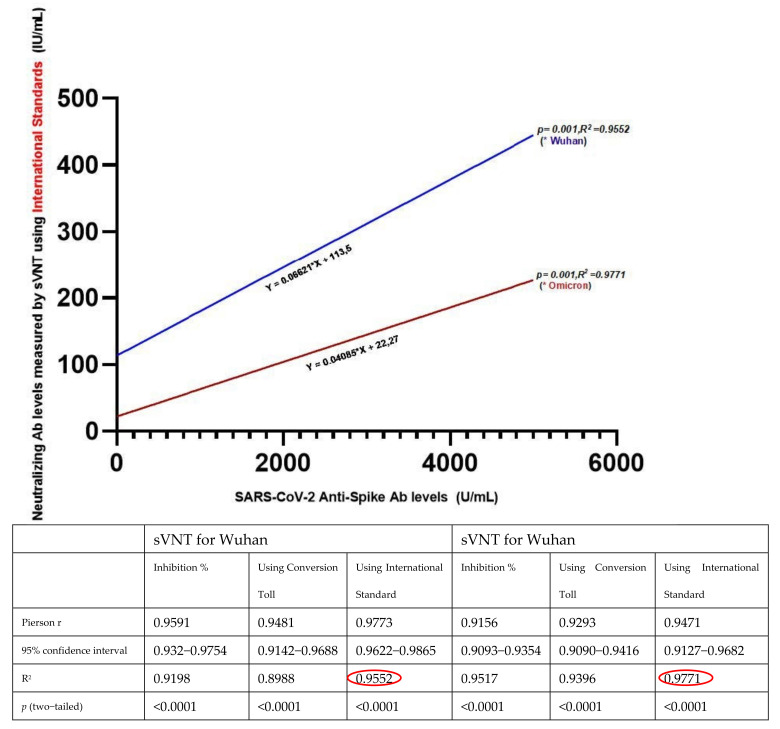
Strong correlation between anti–SARS–CoV–2 spike total Ab (ECLIA) and NAb levels (sVNT) for both Wuhan and Omicron variants, the highest value was obtained when using the international standard (shown in red on the figure). Pierson r, Confidence interval, R^2^, and *p* values of sVNT results obtained using the percentage of signal inhibition, conversion tool, and international standards. (*p* < 0.05 and R^2^ = 0.5–1 were considered statistically significant) (bottom).

**Table 1 vaccines-11-00355-t001:** Demographic data of participants included in the study.

	Previously Infected (n = 201)	Non-Previously Infected (n = 185)	*p* Value
Age (years, mean ±Std)	36.53 ± 8.41	37.59 ± 9.03	0.39
Gender (n)	114 female/87 male	110 female/75 male	0.48
Risky Medical Condition (n)	56 (28.1%)	36 (19.7%)	<0.05 *
Smoking (n)	65 (32.3%)	40 (21.6%)	<0.02 *
BMI (>30) (n)	26 (12.9%)	26 (14%)	<0.05 *
Comorbidities			
➢ Diabetes Mellitus (DM) (n)	4 (1.9%)	4 (2.1%)	0.32
➢ Hypertension (HT) (n)	16 (7.9%)	15 (8.1%)	0.34
➢ Pulmonary disease (n)	9 (4.4%)	1 (0.5%)	<0.05 *
➢ Immune deficiencies (Cancer, auto-immune disease) (n)	4 (1.9%)	0 (0%)	<0.05 *

* *p* < 0.05 is statistically significant.

**Table 2 vaccines-11-00355-t002:** Geometric means of anti-SARS-CoV-2 spike total Ab and neutralizing Ab titers of the study groups in four sampling periods for 12 months (PI = previously infected, NPI = non-previously infected, PIBB = previously infected BioNTech booster receiver, PICB = previously infected CoronaVac booster receiver, PINB = previously infected non booster receiver, NPIBB = non-previously infected BioNTech booster receiver, NPICB = non-previously infected CoronaVac booster receiver, NPINB = non-previously infected non booster receiver groups, n = number of the participants, and 95% CI = 95% confidence interval of GMTs).

Sampling Time	Previously Infected Group (n = 201)			Non–Previously Infected Group (n = 185)			*p* Value
	Anti-SARS-CoV-2 spike total Ab (U/mL)						
(1) 3rd month	275.6 (102.4–1057.9)			60.7 (9.8–76.3)			<0.001 *
	PIBB	PICB	PINB	NPIBB	NPICB	NPINB	
(2) 6th month	11,237 (2246–39,588)	787.2 (271.5–916.3)	234.0 (10.1–627.4)	13,266 (1879–35,352)	759.1 (144.8–831.2)	275.6 (55.6–672.5)	
(3) 9th month	5281 (3732–7739)	475.4 (248.7–734.8)	85.8 (23.1–96.7)	6261 (4568–11,214)	356.2 (224.2–515.9)	64.1 (22.5–82.1)	
(4) 12th month	4138 (3266.7–6394)/ † 12,705 (4874–4216)	91.1 (77.2–103.0)	80.5 (69.9–93.6)	4561 (3822–7411)/ † 13,105 (8634–8103)	86.2 (14.2–101.2)	68.3 (16.5–99.7)	
	Anti–SARS–CoV–2 neutralizing antibodies–NabWuhan, NabOmicron (95% CI) (IU/mL)						
(1) 3rd month	148.5 (58.6–135.4), 98.3 (96.2–101.3)			30.5 (18.6–30.45), 18.8 (14.6–24.2)			<0.001 *
	PIBB	PICB	PINB	NPIBB	NPICB	NPINB	
(2) 6th month	436.8 (426.2–445.1), 145.4 (134.0–156.5)	85.6 (78.8–89.1), 12.8 (11.4–24.5)	NA	422.5 (412.8–433.2), 54.4 (41.26–63.3)	104.5 (97.6–104.0), 11.3 (8.6–27.1)	NA	
(3) 9th month	322.7 (317.5–331.9), 118.3 (106.6–155.2)	50.7 (46.6–57.4), 13.3 (9.6–30.5)	NA	363.6 (17.8–44.0, 95.1 (92.2–111.7)	71.3 (65.1–77.2), 22.5 (20.1–25.6	NA	
(4) 12th month	† 429.3(426.2–444.1), † 210.9 (202.0–223.5)	21.5 (17.7–26.1), 11.2 (6.3–18.6)	NA	† 412.4 (410.3–416.8), † 206.2 (202.5–211.9)	23.6 (19.8–26.5), 10.8 (8.4–13.7)	NA	

** p* < 0.05 is statistically significant. †, Additionally vaccinated HCWs with Biotech as the fourth dose (second BioNTech booster). NA = not applied.

## Data Availability

The data that support the findings of this study are available from the corresponding author upon reasonable request.

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
