# Peer review of "One-Year Post-Vaccination Longitudinal Follow-Up of Quantitative SARS-CoV-2 Anti-Spike Total Antibodies in Health Care Professionals and Evaluation of Correlation with Surrogate Neutralization Test"

_vaccines, 2023, doi:10.3390/vaccines11020355_

Round 1

Reviewer 1 Report

The study has presented the one-year follow-up of the humoral immunity for SARS-CoV-2 of the healthcare worker after boosters with or without BioNTech or CoronaVac. The authors concluded that BioNTech effectively boosted the humoral immunity for SARS-CoV-2 in 6th month.

Here are my contributions to improving the manuscript.

Please add the meaning of HCW (Abstract: lines 28, 107), PI and NPI (lines 37, 38), DM, and HT (Table 1).

Please correct the typo: seveteen (line 213), +Std (Table1, please see line 199)

Table 1: Please add (years) to “Age” and (n) to other specifications, and correct the Age line as Age (years, mean ±Std), 36.53 ±8.41, and 37.59±9.03. 

Do you have any findings about the Anti-SARS-CoV-2 spike total Ab titer and Nab levels of PI group before 1st dose of vaccination? The difference between the PI and NPI after the third month of primary vaccination is expected because of the previous infection. Knowing if Ab levels are lowering despite the vaccination may be helpful.

The authors should address figures more frequently to make the manuscript comprehensive. 

The authors should add a table or tables for the 3rd, 6th, and 12th-month follow-up measurements.

The sentences (lines 278-281) should be placed in line 225.

Author Response

Response to reviewer 1

  • Please add the meaning of HCW (Abstract: lines 28, 107), PI and NPI (lines 37, 38), DM, and HT (Table 1).

health care worker (HCW), previously infected (PI) and non-previously infected (NPI),Diabetes Mellitus (DM) and Hypertension (HT)   ; are added to the text and the table.        

  • Please correct the typo: seveteen (line 213), +Std (Table1, please see line 199)

               One hundred seventeen….., standard deviation (+Std) and Age mean (+Std) are corrected in the text and the table.  

  • Table 1: Please add (years) to “Age” and (n) to other specifications, and correct the Age line as Age (years, mean ±Std), 36.53 ±8.41, and 37.59±9.03.

              Age (years, mean ±Std), 36.53 +8.41, 37.59 +9.03 are corrected and (n) is added on the table to other specifications.

  • Do you have any findings about the Anti-SARS-CoV-2 spike total Ab titer and Nab levels of PI group before 1st dose of vaccination? The difference between the PI and NPI after the third month of primary vaccination is expected because of the previous infection. Knowing if Ab levels are lowering despite the vaccination may be helpful.

Unfortunately, because the study was planned after the vaccination program started in Turkey, in January, 2021, we were unable to obtain serum samples to evaluate Anti-SARS-CoV-2 spike total Abs or NAbs before the first dose of vaccination. There was a period of about one year between the onset of the pandemic and the start of vaccination. COVID-19 exposures of healthcare workers during this time were confirmed based on PCR results and hospital records. The start of the serum sampling 3 months after the primary vaccination with 2 doses of CoronaVAc is described in the text (lines 111 and 124) and the figure 1 (on the left side of the graphical abstract). And it was mentioned that we could not evaluate the pre-vaccination baseline humoral immune response status of the participants, in the limitations.

  • The authors should address figures more frequently to make the manuscript comprehensive.

Added.

  • The authors should add a table or tables for the 3rd, 6th, and 12th-month follow-up measurements.

Table 2 was added.

  • The sentences (lines 278-281) should be placed in line 225.

The replacement of the sentences to the upper lines was done.

Reviewer 2 Report

Tok et al. evaluated the immunogenicity of heterologous or homologous boosters after a two-dose inactivated SARS-CoV-2 vaccine followed by Biontech mRNA or Corona Vac vaccine among HCWs at four different time points during one-year post-vaccination. They measured the humoral responses including anti-SARS-CoV-2 spike-RBD total antibody levels and SARS-CoV-2 neutralization antibodies against the ancestral Wuhan and the Omicron variant in six main groups according to their previous COVID-19 exposure and vaccination status. They found that a heterologous booster of the Biontech mRNA vaccine induced a robust immune response, regardless of the history of previous infection. They also found an increased incidence of COVID-19 in the groups vaccinated with two or three doses of CoronaVac compared to those who vaccinated with Biontech as the booster.

The subject is of great interest at present, but the information presented in the manuscript is not well-structured and clearly stated and some concerns have been raised that need to be addressed.

-          The results should be better organized with deeper explanations to point out the significance of results.

-          Was the first serum sampling 3 months after the first or the second dose of Corona Vac?

-          Please clarify the time of any booster administered.

-          Based on the figure1, please clarify that which subjects are included in the figures 2 and 3 for 9th and 12th month time points and what test, if any, was applied (in the legend captions).

Author Response

Response to reviewer 2

-          The results should be better organized with deeper explanations to point out the significance of results.

       Results reevaluated and organized, figure descriptions detailed. Table 2 was added to better understanding the results, and figures and tables have also been cited more frequently where necessary, making it easier to relate them to the text.

-          Was the first serum sampling 3 months after the first or the second dose of Corona Vac?

      The first serum sampling, showed as the 3rd month sampling on the left side of the figure 1, was performed in June, 3 months after the 2nd CoronaVac vaccine. But it was mentioned once more in the text as follows;

     Participants and Study Design

     HCWs, who presented evidence of the primary vaccination were invited to participate in this prospective cohort study, with a first sampling 3 months after the second dose of CoronaVac, in June 2021 (Figure 1).

 -          Please clarify the time of any booster administered.

      After primary vaccination with 2 doses of CoronaVac, most of the participants in the study received the first booster vaccine as Biontech or CoronaVac in July, while some participants did not (non-booster receiver groups-PINB/NPINB groups). After that, some participants received their second booster vaccine at different intervals (3 months or 6 months after the first booster). In this case, some participants gave serum sample as having received the 2nd booster vaccine in the 9th month sampling, while others gave the serum sample as having received the 2nd booster vaccine in the 12th month sampling. Accordingly, they were classified into different subgroups. All these boosters are summarized in Figure 1. Sampling dates are clarified on Figure 1 and also these dates are added to the text.

-          Based on the figure1, please clarify that which subjects are included in the figures 2 and 3 for 9th and 12th month time points and what test, if any, was applied (in the legend captions).

      In Figure 2, evaluation of anti-SARS-CoV-2 spike total Log10 Ab levels were summarized. Participants of the main arm of the study in 6 groups; those who had or did not have a previous COVID-19 and then who had Biontech or CoronaVac booster after primary vaccination with 2 doses of CoronaVac, and those without booster, were included to the 9th month analyzes (three months after the second sampling). In the 9th month analysis, those who had COVID-19 during this period, those who received a second booster or those who did not receive a booster before and received their first booster during this period were not included. In the 12th month analysis, the arm that received fourth doses of Biontech booster and the main arms remaining on 2nd and 3rd doses of CoronaVac vaccine during this period were included. Participants who had COVID-19 during this period or those who did not receive any booster before and received their first booster during this period were excluded from the 12th month analysis. These small sample groups, which were not included in the analyses, were mentioned anecdotally.

      In Figure 3 evaluation of anti-SARS-CoV-2 N Ab levels were summarized (3a. N AbWuhan levels, 3b NAbOmicron levels). Only the data from 60 selected participants NAbs were compared in four groups who received first and second Biontech or CoronaVac booster, with or without previous COVID-19 (PIBB, PICB, NPIBB and NPICB) at four sampling time-point. Those who did not receive any boosters after two-dose CoronaVac, those who did not receive a 2nd booster yet, in the 12th month sampling, those who received a 3rd booster, or those who had COVID-19 between the sampling periods were not included in the sVNT.

     These explanations were added to the figure 2 and 3 descriptions.

Round 2

Reviewer 1 Report

I thank the authors for the corrections and the addition of table 2. Here are my contributions to Table 2. The legend should be at the top. Additionally, the table displays two pages. And finally, the authors should check the punto and font of the text.  

Author Response

Response to reviewer 1

  • Table 2. The legend should be at the top. Additionally, the table displays two pages. And finally, the authors should check the punto and font of the text.  

As seen below; I took the table legend at the top, rearranged the column widths and reduced the font size to fit on a single page, and finally checked the font type and made it 'Palatino Linotype' as in the template.

Table 2. Geometric means of anti-SARS-CoV-2 spike total Ab and neutralizing Ab titers of the study grups in four sampling periods for 12 months (PI= previously infected, NPI= non-previously infected, PIBB= previously infected-Biontech booster receiver, PICB= previously infected-CoronaVac booster receiver, PINB= previously infected-non booster receiver, NPIBB= non-previously infected-Biontech booster receiver, NPICB= non-previously infected-CoronaVac booster receiver, NPINB= non-previously infected-non booster receiver groups, n=number of the participants,  95% CI =95% confidence interval of GMTs).

Sampling Time

Previously Infected Group

(n=201)

Non-Previously Infected Group (n=185)

p value

Anti-SARS-CoV-2 spike total Ab (U/mL)

(1) 3rd month

275.6 (102.4-1057.9)

60.7 (9.8-76.3)

<0.001*

PIBB

PICB

PINB

NPIBB

NPICB

NPINB

(2) 6th month

11237

(2246-39588)

787.2

(271.5-916.3)

234.0

(10.1-627.4)

13266

(1879-35352)

759.1

(144.8-831.2)

275.6

(55.6-672.5)

(3) 9th month

5281

(3732-7739)

475.4

(248.7-734.8)

85.8

23.1-96.7)

6261

(4568-11214)

356.2

(224.2-515.9)

64.1

(22.5-82.1)

(4) 12th month

4138 (3266.7-6394) /

†12705 (4874-4216)

91.1

(77.2-103.0)

80.5

(69.9-93.6)

4561 (3822-7411) /

†13105 (8634-8103)

86.2

(14.2-101.2)

68.3

(16.5-99.7)

Anti-SARS-CoV-2 neutralizing antibodies-NAbWuhan, NAbOmicron (95% CI) (IU/mL)

(1) 3rd month

148.5 (58.6-135.4), 98.3 (96.2-101.3)

30.5 (18.6-30.45), 18.8 (14.6-24.2)

<0.001*

PIBB

PICB

PINB

NPIBB

NPICB

NPINB

(2) 6th month

436.8 (426.2-445.1),

145.4 (134.0-156.5)

85.6 (78.8-89.1),

12.8 (11.4-24.5)

NA

422.5 (412.8-433.2), 54.4 (41.26-63.3)

104.5 (97.6-104.0),

11.3 (8.6-27.1)

NA

(3) 9th month

322.7 (317.5-331.9),

118.3 (106.6-155.2)

50.7 (46.6-57.4),

 13.3 (9.6-30.5)

NA

363.6 (17.8-44.0,

95.1 (92.2-111.7)

71.3 (65.1-77.2),

22.5 (20.1-25.6

NA

(4) 12th month

† 429.3(426.2-444.1),

† 210.9 (202.0-223.5)

21.5 (17.7-26.1),

11.2 (6.3-18.6)

NA

† 412.4 (410.3-416.8),

† 206.2 (202.5-211.9)

23.6 (19.8-26.5),

10.8 (8.4-13.7)

NA

*p <0.05 is statistically significant

†Additionally vaccinated HCWs with Biotech as the fourth dose (second Biontech booster)

 NA=not applicated

Reviewer 2 Report

The manuscript is interesting on an important topic. The research hypothesis is consistent and the experiments are well designed.

Author Response

No spesific revision was  requested.
